# Debris Flow Prediction Based on the Fast Multiple Principal Component Extraction and Optimized Broad Learning

**Genqi Xu** [1], **Xin-E Yan** [1,*], **Ning Cao** [1], **Jing Ma** [2,3], **Guokun Xie** [1] **and Lu Li** [4]

1 Xi'an Traffic Engineering Institute, Xi'an 710300, China
2 State Key Laboratory Base of Eco-Hydraulic Engineering in Arid Area, Xi'an 710048, China
3 China Gezhouba Dam Group Third Engineering Co., Ltd., Xi'an 710065, China
4 Xi'an Siyuan University, Xi'an 710038, China
* Correspondence: yanxine@126.com; Tel.: +86-13310945637

**Abstract:** In the current research of debris flow geological disaster prediction, determining reasonable disaster-inducing factors and ensuring the accuracy and rapidity of the prediction model are considered vital issues, and also, essential foundations for disaster early warning and disaster prevention and mitigation. Aiming at the problems of low prediction accuracy and long prediction time in the current debris flow research, firstly, six debris flow impact factors were selected relying on the fast multiple principal component extraction (FMPCE) algorithm, including rainfall, slope gradient, gully bed gradient, relative height difference, soil moisture content and pore water pressure. Next, based on the broad learning (BL) algorithm, the debris flow prediction model based on FMPCE and the optimized BL is established with the input of debris flow-inducing factors and the output of debris flow probability. Then the model is optimized using matrix stochastic approximate singular value decomposition (SVD), and the debris flow disaster prediction model, based on SVDBL, is constructed. The prediction results of the optimized model are compared with those of the gradient descent optimized the BP neural network model (GD-BP), Support Vector Machines model (SVM) based on grid search and BL model. The results show that the accuracy of SVDBL is 7.5% higher than that of GD-BP, 3% higher than that of SVM and 0.5% higher than that of BL. The RMSE sum of SVDBL was 0.05870, 0.0478 and 0.0227 less than that of GD-BPSVM and BL, respectively; the MAPE sum of SVDBL was 1.95%, 1.66% and 0.49% less than that of GD-BPSVM and BL; the AUC values of SVDBL were 12.75%, 7.64% and 2.79% higher than those of the above three models, respectively. In addition, the input dataset is expanded to compare the training time of each model. The simulation results show that the prediction accuracy of this model is the highest and the training time is the shortest after the dataset is expanded. This study shows that the BL can be used for debris flow prediction, and can also provide references for disaster early warning and prevention.

**Keywords:** fast multiple principal components extraction; broad learning; debris flow; prediction model; matrix random approximation singular value decomposition

## 1. Introduction

Debris flow is one of the main geological disasters that is widely distributed in mountainous areas all over the world and threatens the lives and property of people all over the world [1] Large-scale mudslides have frequently occurred all over the world, and thousands of people have lost their lives [2]; debris flows of various scales can occur nearly 100,000 times a year all over the world. Since 2014, the United States and Japan, the two most technologically developed countries in the world, have successively experienced major geological disasters causing deaths due to mudslides [3].

Especially in China, the natural environment is complex and changeable, and the geological structure activities in mountainous areas are vital. Mountains, plateaus and hills account for about two-thirds of the land area, and the terrain elevation difference is

pronounced, the western region high and the eastern region low, forming three steps [3]. Water erosion and scouring in this terrain will be severe and easily cause flooding, soil erosion and geological disasters on the slopes [4]. Most of China has a typical continental monsoon climate; the rainfall is too concentrated and clamped by the Indian and Pacific plates. Under the background of the neotectonic movement, the inland areas of China have well-developed fault structures and a high frequency of seismic activity, which leads to the occurrence frequency, distribution range and damage scale of debris flow in the world [5].

In recent years, the debris flow has been more severe with the increase in unreasonable construction activities such as the random transformation of mountains and forests. The frequent occurrence of debris flow has attracted the attention of researchers and related workers. To reduce the harmful degree of debris flow, the relevant departments have strengthened the investigation and prevention work and investigated and explored the areas prone to debris flow [6]. Debris flow prediction has become a frontier research topic in the field of natural disasters. Providing an effective prediction method for debris flow has become a critical point of disaster prevention and mitigation [7].

With the rapid development of GIS technology, the overall idea of spatial prediction of debris flow has been formed, which is to establish the disaster-inducing factor system of debris flow according to the characteristics of debris flow formation, and use specific prediction methods to comprehensively analyze various disaster-inducing factors to form the prediction and forecast of debris flow [8–10]. Although the overall idea of the current research is relatively unified, the prediction models are different. The existing debris flow prediction models can be divided into two categories, including the deterministic model [11] considering the internal physical–mechanical mechanism of debris flow formation, and the non-deterministic model based on statistical principle [12], fuzzy theory [13], pattern recognition [14] and various composite models [15–17].

Specifically, in terms of deterministic model research, Liu et al. [18] used GIS technology to evaluate the water–soil mixture density RHO in each debris flow watershed in real time through the distributed hydrological model and unsaturated soil limit equilibrium method. They used the RHO to determine the probability of debris flow in each debris flow watershed in the region. Yu et al. [19] found a refined debris flow prediction model based on gully bed width and particle size. Banihabib et al. [20] ran experiments on different disaster-inducing factors of debris flow, and derived an empirical formula for determining the critical value of debris flow. Wei et al. [21] used the moving least squares (MLS) to predict the debris flow velocity. Imaizumi et al. [22] predicted the onset and runoff characteristics of debris flow gushing based on the hydrological distribution of the basin. In the study based on the statistical principle model, Dal Pont et al. [23] established a debris flow prediction model based on the partial correlation coefficient of debris flow disaster-inducing factors to determine the probability of debris flow. Giannecchini et al. [24] used logistic regression to predict debris flow around a quarry in the Alps, Italy. Based on the principle of the efficacy coefficient method, Meng et al. [25] used the improved analytic hierarchy process to calculate the weight coefficient of evaluation factors and established a debris flow prediction and early-warning model based on the comprehensive analysis of the meteorological and geological environmental factors affecting the occurrence of debris flow disasters. Zhang et al. [26] established a debris flow prediction model based on a watershed-scale water–soil coupling mechanism through distributed hydrological model (GBHM) and unsaturated soil instability identification model according to the observation data of CAS Dongchuan debris flow observation and research station located in Jiangjia Gully. Liang et al. [27] proposed a comprehensive prediction model of mine debris flow based on a variable weight cloud model. Qiao et al. [28] suggested a debris flow prediction method based on a rainfall-unstable soil coupling mechanism (R-USCM) by introducing the Scoops3D model. Benhabib et al. [29] proposed a Bayesian network (BN) model combined with the K-means clustering method to predict debris flows in the northern basin of Iran. Wu et al. [30] expected the occurrence of debris flows by studying the rainfall duration intensity threshold. Kern et al. [31] proposed a probability prediction

method for post-volcanic debris flow based on machine learning. In the study based on the fuzzy theory model, Xiao LM [32] et al. used fuzzy theory to analyze, evaluate and warn about debris flow disasters. Su et al. [33] established a comprehensive debris flow prediction model based on soil flow, rock flow, typhoon and rainfall history information using the K-nearest neighbor algorithm. In the research in pattern recognition models, Hirschberg et al. [34] predicted debris flow in alpine valleys based on a decision tree algorithm. Wang et al. [35] proposed a new method of debris flow early warning based on the optimized feature self-organizing feature mapping network. Cao et al. [36] raised a debris flow prediction method based on the GSA-RBF model. Zhang et al. [37] proposed a debris flow prediction model based on the relevance vector machine. Lee et al. [38] predicted rainfall debris flow relies on the artificial neural network (ANN). Mei et al. [39] established the support vector machines (SVM) model for debris flow prediction. In other model research fields, Hu et al. [40] studied debris flow infrasound early warning based on wavelet transform. Nikolopoulos EI et al. [41] evaluated the debris flow prediction model using random forest. Min Y et al. [42] considering the internal and external factors affecting landslide and debris flow, combined with quantitative precipitation forecasting business, established a landslide and debris flow disaster forecasting and early warning model in Yunnan Province. The probability and scale of debris flow are predicted by Kean et al. [43] on the historical distribution of fire and precipitation frequencies. Cao [36] combined a gravity search algorithm and an RBF neural network to construct a debris flow prediction model.

Although some achievements have been made in the above research, there are still some imperfections. For example, only part of the factors can be considered in an established deterministic model with significant limitations. Logistic regression belongs to the generalized linear regression model. Although multiple indicators can be synthesized in logistic regression, it is still greatly affected by multicollinearity. The calculation complexity of the fuzzy system is enormous, and it is too subjective to determine the weight vector of the debris flow index. Although ANN can overcome this defect, the model quickly falls into the local minimum value in the training process of artificial neural networks. Therefore, the accuracy of this method is not high. The AHP relies on the support of an expert system. It often has a large-scale workload and many data dimensions, which make it easy to produce dimensional curses. RBF neural network can be complementary with numerical simulation and empirical formula method. Still, it is a lack of correlation analysis between indicators, which may cause information superposition and also produce dimension curses.

To solve the problems of subjective selection of disaster impact factors, information redundancy in input data, insufficient prediction accuracy and poor real-time prediction in the current debris flow prediction research in the above methods, the debris flow probability prediction model is established by combining the fast multiple principal component extraction (FMPCE) [44] and broad learning (BL) [45]. The FMPCE algorithm extracted the disaster-inducing factors to replace the initial high latitude shadow, which aims to improve the problem caused by traditional methods. The BL algorithm is used to predict the probability of debris flow, which aims to compensate for the long training time driven by the need to calculate a large number of hidden layer weights in deep learning, and improve the online update ability of the model. The SVDBL model is established using matrix random approximation singular value decomposition (SVD) to optimize the model. This can solve the problem of input matrix structure redundancy caused by poor model initialization.

Given this, this study aims to seek a rapid and accurate debris flow disaster prediction method, mainly for the occurrence of debris flow disaster probability forecast. Managers and decision-makers can formulate different levels of countermeasures or emergency measures based on the forecast results to help local people move people and property ahead of time, thus effectively reducing the harm caused by disasters to local people. In the first step, sensors and field surveys obtain the initial impact factors. In the second step, the FMPCE algorithm is used to screen out the impact factors that have a more significant

contribution rate to the occurrence of debris flow. In the third step, the main impact factors are divided into a training set and a verification set. In the fourth step, the data of the training set is input into the BL model for training. The BL model is optimized in the fifth step by the matrix stochastic approximate singular value decomposition algorithm. In the sixth step, the optimized SVDBL model is tested by the data in the verification set.

## 2. Overview of the Study Area

### 2.1. Topography and Geomorphology

The study area is located in the Qinling Mountains and Shanyang County southeast of Shangluo. It is adjacent to Shangzhou in the east, Zhashui in the west, Hubei Province in the south and Shangzhouof the dimensional curse district in the north. Shanyang County borders Liuling in the north, Yunling in the south and Juan Ling in the middle. The rivers flowing through include the Qianqian and Xiejia rivers, which belong to the branches of the Yangtze and Hanjiang rivers. The river length is 40 km and 80 km, respectively, and the drainage area is 600 km$^2$ and 2400 km$^2$. Its altitude range is 800~1500 m, located under the Qinling Mountains, with many mountains and gullies, belonging to the middle and low mountain terrain. The mountain's amount of soil and stone is as high as 1,800,000 km$^2$ accounting for more than 80% of the area, forming a canyon area with significant terrain differences. At the same time, it is located in the middle of the two rivers, with many small river basins and abundant water resources. In summer and autumn, the rainfall is more, and the temperature changes significantly, which is easy to cause loose soil. The region is rich in minerals and the intensity of human activities increases the debris flow safety points.

### 2.2. Distribution and Law of Debris Flow

There are various types of geological disasters, and their occurrence is affected by complex topography, meteorology and weathering. Still, their affair spatial and temporal patterns can be studied by data characteristics. According to the data analysis of regional geological disasters in the past ten years, debris flow mainly occurs near rivers and mountainous areas. There are as many as 180 gullies, most of which are strip-shaped and some are point-shaped, accounting for 16% of the total number of disasters. The spatial distribution of geological hazards in the study area is shown in Figure 1.

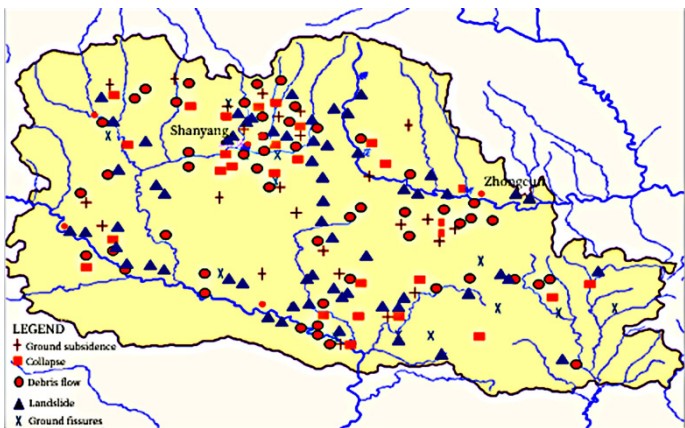

**Figure 1.** Spatial distribution of geological hazards in the study area.

Based on the statistical analysis of the characteristics of debris flow in Shanyang County in recent years, it is concluded that most of the disasters are related to the intensity and season of rainfall, and some of them are related to the excessive exploitation of human beings. The power of precipitation affects the occurrence of disasters. However, rainfall is related to seasons, so there is a corresponding relationship in time. The study area has a semi-humid mountain climate, and the rain in summer is relatively affluent, especially some continuous heavy rainfall that provides power for geological disasters. According to

statistics, a large amount of rainfall accumulated from May to October in this area, which is easy to cause debris flow.

The study of debris flow geological hazard is complex nonlinear system research, which has the characteristics of wide regional distribution, complex monitoring and modeling, a large amount of data and instability. Only by analyzing and studying the geological and geomorphological laws and a large amount of data, laying a sound data foundation, selecting appropriate disaster influencing factors, and establishing a disaster prediction model can debris flow be prevented more effectively.

## 3. Theories and Method

### 3.1. FMPCE

There are many factors inducing debris flow, including rainfall, soil moisture content, pore water pressure, watershed development degree, watershed integrity coefficient, vegetation coverage, slope gradient, lithologic, watershed area, relative height difference, erosion and deposition amplitude, gully bed gradient, loose material reserves along the gully, adverse geological phenomena, new structure influence, recharge section length ratio and human activities, which make it challenging to train the prediction model. In this paper, the FMPCE is used to screen the influencing factors of debris flow prediction. The specific methods are as follows.

For the linear neural network model,

$$y(k) = W^T(k)x(k) \tag{1}$$

where $y(k) \in R^{r \times 1}$ and $x(k) \in R^{r \times 1}$ are the neural network output and input, respectively, and $W(k) \in R^{n \times r}$ is the weight matrix of the neural network. Among them, the $n$ and $r$ are the input vector dimension and the dimensions of all extracted principal components, respectively. Making the input autocorrelation matrix $R = E\left[x_k x_k^T\right]$, $R$ needs to be a symmetric positive matrix. The eigenvalue of $R$ is denoted by $\lambda_i$ while the $u_i$ is the eigenvector belonging to the eigenvalue $\lambda_i$, $i = 1, 2, \ldots, n$, and $\lambda_i > 0$. Then, the eigenvalue decomposition of $R$ can be performed by Equation (2).

$$R = U \cap U^T \tag{2}$$

where $U = [u_1, u_2, \ldots, u_n]$, $\Lambda = \text{diag}\{\lambda_1, \lambda_2, \ldots, \lambda_n\}$, and the eigenvalues need to meet the requirements of Equation (3).

$$\lambda_1 > \lambda_2 > \ldots > \lambda_r > \ldots > \lambda_n > 0 \tag{3}$$

The eigenvectors belonging to these r eigenvalues are the first r principal components of the matrix $R$, and the space generated by these main components is called the principal subspace. FMPCE is to find the appropriate weight matrix iterative update equation, so that the weight matrix can converge to the first $r$ main components of matrix $R$. The algorithm form is shown in Equation (4).

$$W(k+1) = W(k) + \eta W(k)C(k) + (E(k)A^2 - F(k)A) \tag{4}$$

where the $A$ is a $r \times r$ diagonal matrix, The diagonal elements are $a_1 > a_2 > \cdots > a_n > 0$, and the $\eta$ is the learning rate. $C(k) = W(k)((W(k)^T W(k))^{-1} - I)$ is a non-second-order matrix, the introduction of $C$ can not only solve the instability problem of the algorithm, but also improve the convergence speed of the algorithm [15]. $E(k) = RW(k)W^T W(k)$, $F(k) = W(k)AW^T(k)RW(k)$.

In practical applications, the autocorrelation matrix is not known, it needs to be estimated by Equation (5).

$$\widehat{R} = (1 - 1/k)\alpha \widehat{R}(k-1) + (x_k x_k^T)/k \tag{5}$$

where $\alpha$ is the forgetting factor, which needs to meet the requirements of $0 < \alpha < 1$, Obviously, when $k \to \infty$, the matrix $\widehat{R}(k) \to R$. First, the autocorrelation matrix is estimated by using Equation (5). Then, principal input components can be extracted by using Equation (1) and Equation (4). The FMPCE algorithm flow is shown in Figure 2.

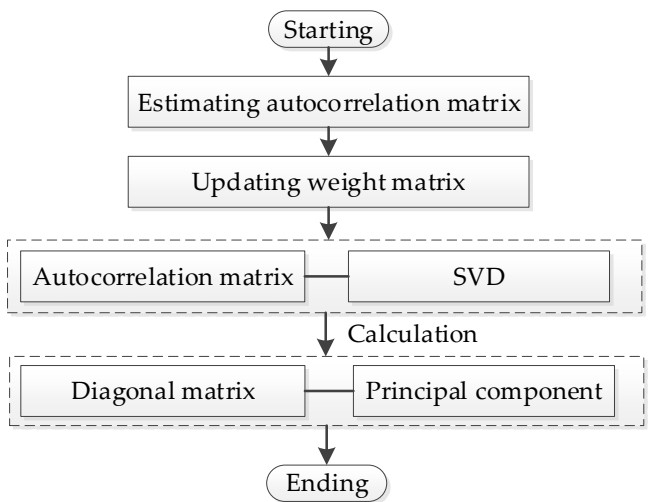

**Figure 2.** The implementation process of FMPCE.

### 3.2. Broad Learning

First, the input data is mapped into a feature node matrix through the BL, and then the enhanced node matrix is formed by enhanced transforming based on the previous. Finally, the weight matrix between the hidden layer and output can be solved by the pseudo-inverse method with the input of feature mapping nodes and enhancement nodes. Throughout the process, only the connection between the hidden layer and the output is updated, which makes the training process simple. Moreover, once the number of feature mapping and enhanced nodes cannot meet the required accuracy, the BL can be quickly retrained using incremental learning. The specific process is as follows.

#### 3.2.1. Initial Structure

The structure of the BL is shown in Figure 3.

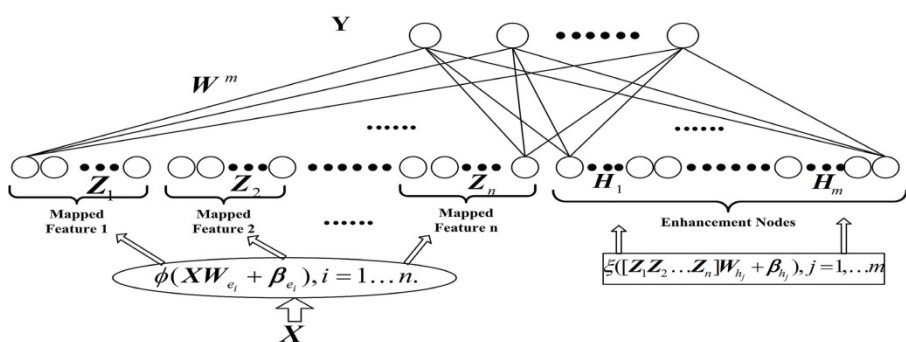

**Figure 3.** The initial structure of the BL model.

The $\Phi_i(XW_{ei} + \beta_{ei})$ denotes the $i$th feature mapping $Z_i$, $\xi_j(Z^iW_{h_j} + \beta_{h_j})$ represents the $j$th enhanced node $H_j$, where the $W_{ei}$ is Random initial weights, $Z^i = [Z_1, Z_2, \dots, Z_i]$, $H^i = [H_1, H_2, \dots, H_i]$. For simplicity, the subscript of $\Phi_i$ and $\xi_j$ is omitted later.

For $n$ feature maps, each mapping node generates $k$ enhanced nodes, expressed as Equation (6)

$$Z_i = \Phi_i(XW_{ei} + \beta_{ei}), i = 1, 2, \dots, n \tag{6}$$

Therefore the breadth learning model is represented as Equation (7).

$$Y = \left[Z_1, \ldots, Z_n \middle| \xi\left(Z^n W_{h_1} + \beta_{h_1}\right), \ldots, \xi\left(Z^n W_{h_m} + \beta_{h_m}\right)\right] W^m = [Z^n | H^m] W^m \tag{7}$$

where $W^m = [Z^n | H^m]^+ Y$ is the connection weight, $[Z^n | H^m]^+$ can be obtained by Equation (8) (pseudo-inverse ridge regression approximation algorithm).

$$A^+ = \lim_{\lambda \to 0} \left(\lambda I + AA^T\right)^{-1} A^T Y \tag{8}$$

### 3.2.2. Updated Structure

Assuming that the number of enhancement nodes inserted is, and $A^m = [Z^n | H^m]$, $A^{m+1} = \left[A^m | H^{m+1}\right]$. According to the dynamic update algorithm of RVFLNN, a new pseudo-inverse of the input matrix $A^{m+1}$ can be obtained through Equation (9)

$$A^{m+1} = \begin{bmatrix} (A^m)^+ - DB^T \\ B^T \end{bmatrix} \tag{9}$$

The new weights $W^{m+1}$ can be calculated by Equation (10).

$$W^{m+1} = \begin{bmatrix} W^m - DB^T Y \\ B^T Y \end{bmatrix} \tag{10}$$

All the above pseudo-inverse matrices are calculated by Equation (8). That means enhanced nodes do not need to figure all in this algorithm. Only the pseudo-inverse of the newly inserted augmentation node needs to be calculated so that it enables fast incremental learning.

Similarly, new feature mapping nodes and new input data can be updated. The Updated structure of the BL model diagram is shown in Figure 4.

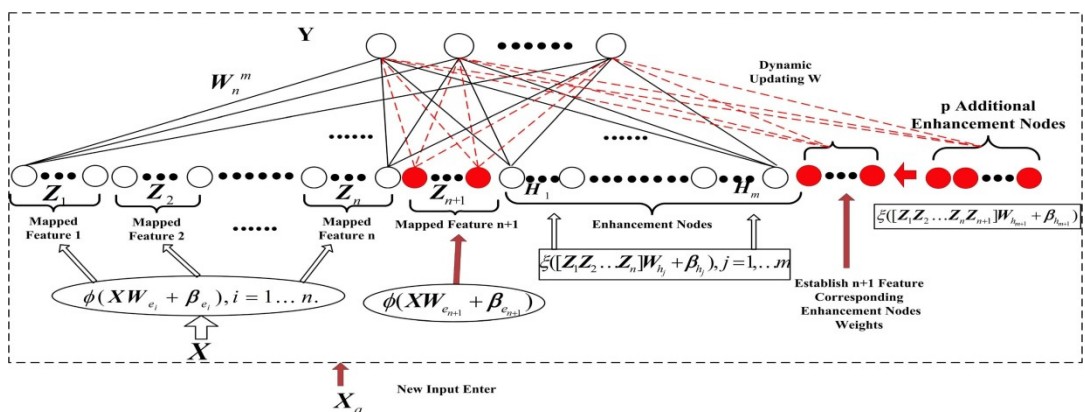

**Figure 4.** Updated structure of the BL model.

### 3.3. Improved Broad Learning

As the initial parameters of the model are randomly generated during learning, the matrix structure of the hidden layer may be redundant due to the improper setting of the initial parameters. The usual way is to optimize the matrix structure with the help of low-rank approximation [46]. Here, the matrix random approximation SVD is used to optimize the above model, and the specific process is described below.

$$Y = [Z_1, \ldots, Z_n] W_n^0 \tag{11}$$

where $A_n^0 = [Z_1, \ldots, Z_n]$ and $Y = A_n^0 W_n^0$.

The purpose of matrix decomposition is to reduce the number of nodes to simplify the calculation. Matrix $Z_i$ is decomposed by using matrix random approximation SVD. A

matrix $Q$ is Constructed by a set of orthogonal bases of column space of the matrix $Z_i$, and where $Z_i \approx QQ^*Z_i$, $Q$ becomes an approximate submatrix of $Z_i$, $QQ^*Z_i$ is the low-rank approximation of the subspace formed by $Z_i$. The approximate matrix of $Z_i$ can be easily found according to $Q$. Then the singular value decomposition of $Z_i$ can be obtained by the approximate matrix of $Z_i$.

$$Z_i = U_{Z_i} \sum_{Z_i} V_{Z_i}^T = Z_i^P + Z_i^Q \tag{12}$$

$$Z_i^P V_i^P = Z_i V_{Z_i}^P \tag{13}$$

Assuming that $W_n^0 = [W_{Z_1}^{\{0,n\}} | \cdots | W_{Z_n}^{\{0,n\}}]^T$, the calculation is shown in Equation (8).

$$Y = A_n^0 W_n^0 = [Z_1, \ldots, Z_n] W_n^0 = \left[ Z_1 V_{Z_1}^P, \ldots, Z_n V_{Z_n}^P \right] \begin{bmatrix} V_{Z_1}^{P\,T} W_{Z_1}^{\{0,n\}} \\ \cdots \\ V_{Z_n}^{P\,T} W_{Z_n}^{\{0,n\}} \end{bmatrix} = A_F^{\{0,n\}} W_F^{\{0,n\}} \tag{14}$$

The model can be defined as $Y = A_F^{\{0,n\}} W_F^{\{0,n\}}$, where $W_F^{\{0,n\}} = \left( A_F^{\{0,n\}} \right)^+ Y$.

The calculation is shown in Equation (8) when the number of inserted enhanced nodes is $p$,

$$\left( A_F^{\{m+1,n\}} \right)^+ = \begin{bmatrix} \left( A_F^{\{m,n\}} \right)^+ - DB \\ B^T \end{bmatrix} \tag{15}$$

$$W_F^{\{m+1,n\}} = \begin{bmatrix} W_F^{\{m,n\}} - DB^T Y \\ B^T Y \end{bmatrix} \tag{16}$$

Finally, smaller singular values still need to be removed for further optimization, and the final result is shown in Equation (17).

$$A_F^{\{m,n\}} = U_F \sum_F V_F^T = U_F \cdot \left[ \sum_F^P \middle| \sum_F^Q \right] \cdot \left[ V_F^P \middle| V_F^Q \right]^T = A_F^{\{m,n\}\ P} + A_F^{\{m,n\}\ Q} \tag{17}$$

Making $A_F = A_F^{\{m,n\}} V_F^P$, then $Y = A_F W_F$, where $W_F = A_F^+ Y$.

## 4. Modeling Process and Result Analysis

### 4.1. Establishment of Debris Flow Probability Prediction Model

To predict the probability of debris flow objectively and accurately, a debris flow prediction model is proposed in this paper, as shown in Figure 5. The structure of the model includes four modules, namely, the data module, technical module, algorithm module and forecasting module. The main functions of each module are as follows.

For the data module, this module is mainly responsible for the historical data of debris flow, including data collection, storage, update, extraction and other operations. Considering that the factors inducing debris flow are constantly changing with the change in environment, the available module is adopted to update and maintain the data in real time.

For the technology module, due to numerous factors affecting debris flow and huge data, some data have no apparent influence on the forecast results, so it is necessary to screen the data. The FMPCE algorithm is adopted in this module to achieve the above purpose.

For the algorithm module, the influencing factors of debris flow are very complex, and often random and fuzzy. To compensate for the slow speed of the existing methods, the BL algorithm is used in this module to train the data, which can significantly reduce the training time. It has obvious advantages in updating the model online at the same time.

For the prediction module, we make the training sample $X = \{x1, x2, x3, x4, x5, x6\}$, where the $x1$, $x2$, $x3$, $x4$, $x5$, and $x6$ is the rainfall, the slope gradient, the gully bed gradient, the relative height difference, the soil moisture content, and the pore water pressure, respectively. The output $Y$ is the probability of debris flow.

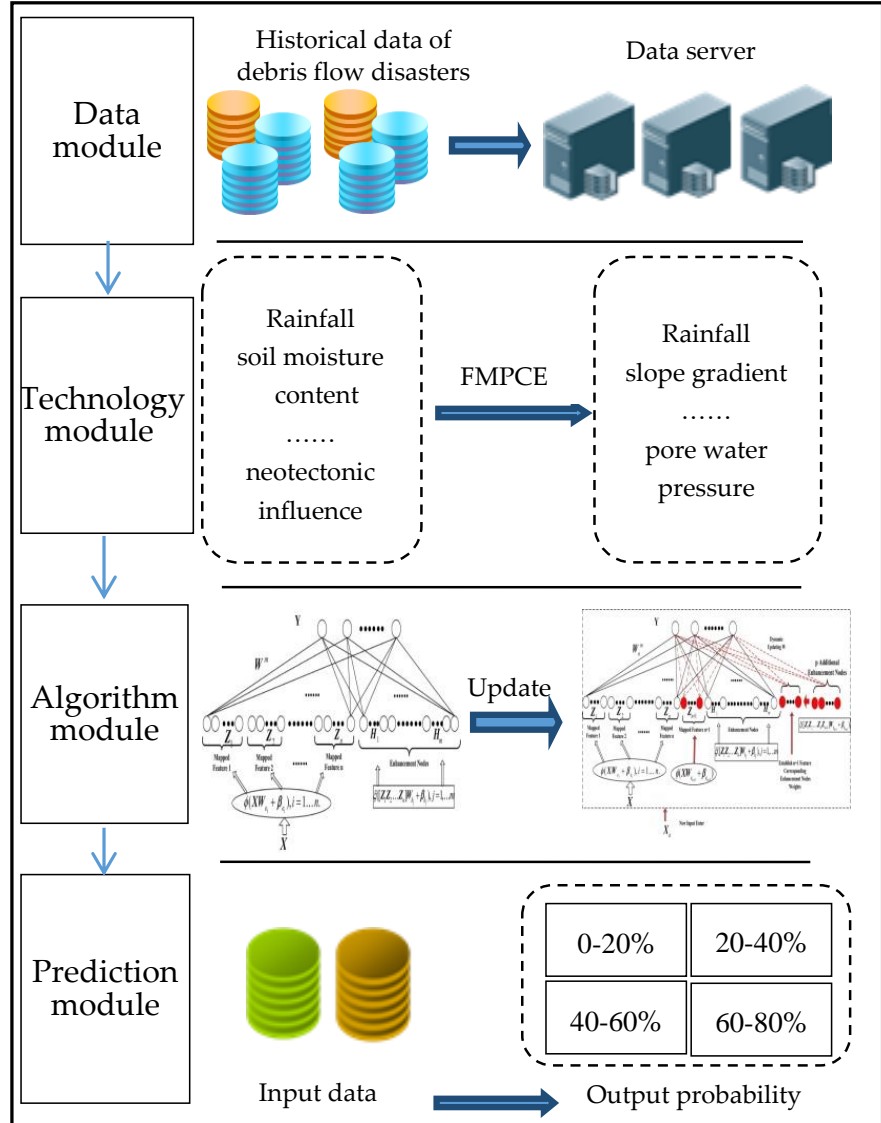

**Figure 5.** Debris flow prediction model.

The initial impact factors obtained from the data module are input into the technology module. Based on the FMPCE algorithm of the technology module, the main impact factors can be quickly screened. After screening, the sample data of main influencing factors are divided into a training set and a verification set. After the BL calculation and SVD optimization of the algorithm module, the probability of debris flow occurrence is predicted accurately and quickly by the prediction module. Finally, the verification set data is directly input into the trained model to verify the performance of the model The specific research scheme is shown in Figure 6.

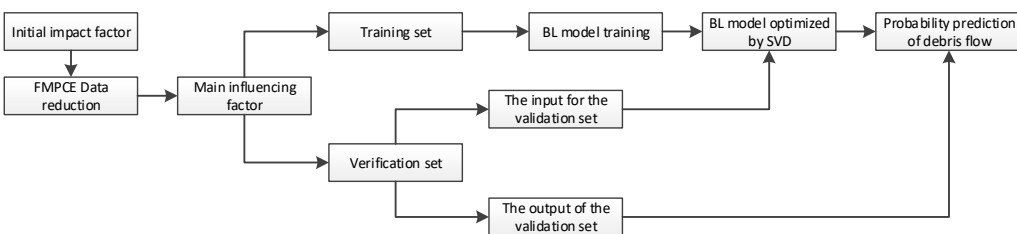

**Figure 6.** Research scheme.

The output $Y$ is the probability of debris flow.

---

**Algorithm 1** Debris flow prediction based on the SVDBL algorithm

---

　　Input: Training sample $(X, Y)$ and actual input $Y$

　　Output: Debris flow probability $\widehat{Y}$

1　**for** i $= 0; i \leq$ n **do**
2　　　Initialize $W_{ei}$ and $\beta_{ei}$
3　　　Calculate $Z_i = \Phi_i(XW_{ei} + \beta_{ei})$
4　　　Calculate $V_{Z_1}^P$ by SVD
5　**end**
6　**for** $j = 0$ ; $j \leq m$ **do**
7　　　Initialize $W_{h_j}$ and $\beta_{h_j}$
8　　　Calculate $H_j = \xi\left(\left[Z_1 V_{Z_1}^P, \ldots, Z_n V_{Z_n}^P\right] W_{h_j} + \beta_{h_j}\right)$
9　　　Calculate $V_{H_1}^P$ by SVD
10　**end**
11 Let $A^{\{m,n\}} = \left[Z_1 V_{Z_1}^P, \ldots, Z_n V_{Z_n}^P \,|\, H_1 V_{H_1}^P, \ldots, H_m V_{H_m}^P\right]$
12 Calculate $\left(A^{\{m,n\}}\right)^+$ by Equation (8)
13 **while** the training accuracy does not meet the requirements **do**
14　　　Initialize $W_{h_{m+1}}$ and $\beta_{h_{m+1}}$
15　　　Calculate $H_{m+1} = \xi\left(\left[Z_1 V_{Z_1}^P, \ldots, Z_n V_{Z_n}^P\right] W_{h_{m+1}} + \beta_{h_{m+1}}\right)$; Update $A^{m+1}$
16　　　Calculate $V_{H_{m+1}}^P$ by SVD
17　　　Update $A^{\{m+1,n\}}$
18　　　Calculate $\left(A_F^{\{m+1,n\}}\right)^+$ and $W_F^{\{m+1,n\}}$ by Equations (22) and (23)
19　　　$m = m + 1$
20 **end**
21 Calculate $V_F^P$ by SVD
22 Calculate $A_F = A_F^{\{m+1,n\}} V_F^P$
23 Calculate $A_F^+$ and $W_F = A_F^+ Y$ by Equation (8)
24　Let $W = W_F$
25　Calculation of debris flow probability $\widehat{Y} = \widehat{X} W$

---

### 4.2. Determine the Influencing Factors and Data Sources

According to the Specification of Geological Investigation for Debris Flow Stabilization (T/GATHP 006-2018), referring to the literature and combined with the field survey and field investigation data, 16 influencing factors are selected through the analysis of the environmental background and development characteristics of the debris flow in the study area (The details are in Table 1). The FMPCE algorithm in the technical module is used to calculate the eigenvalue of each influencing factor (Figure 7) and the contribution rate of each factor (Figure 8), and then 1 to 16 factors are selected in turn according to the contribution rate to train and verify the model in the algorithm module (Figure 9). Finally, the prediction module outputs the results According to the descending order of contribution rate, the influence factors of different dimensions are selected in turn to compare the forecast results, and finally, six influence factors are selected.

It is evident from Figure 9 that with the increase in the dimension of debris flow impact factors, the prediction accuracy of the model is gradually improved. When the number of dimensions reaches six, the model's prediction accuracy is stable. There is no apparent change in the model's prediction accuracy with the increasing number of impact factors. Therefore, the dimension of impact factors is selected as six. According to the FMPCE algorithm mentioned above, the principal components are extracted. The influencing factors finally extracted include rainfall, slope gradient, gully bed gradient, relative height difference, soil moisture content and pore water pressure. Through the study of the changes in correlation parameters of debris flow in multiple areas [47–50],

the relationship between each influencing factor and the probability of debris flow and its representation significance are summarized, as shown in Table 2.

**Table 1.** Influence factors list.

| Number | Influence Factors |
|--------|-------------------|
| 1 | Rainfall |
| 2 | Soil moisture content |
| 3 | Pore water pressure |
| 4 | Watershed development degree |
| 5 | Watershed integrity coefficient |
| 6 | Vegetation coverage |
| 7 | Slope gradient |
| 8 | Lithologic |
| 9 | Watershed area |
| 10 | Relative height difference |
| 11 | Erosion and deposition amplitude |
| 12 | Gully bed gradient |
| 13 | Recharge section length ratio |
| 14 | Loose material reserves along the gully |
| 15 | Adverse geologic phenomena |
| 16 | New structure influence |

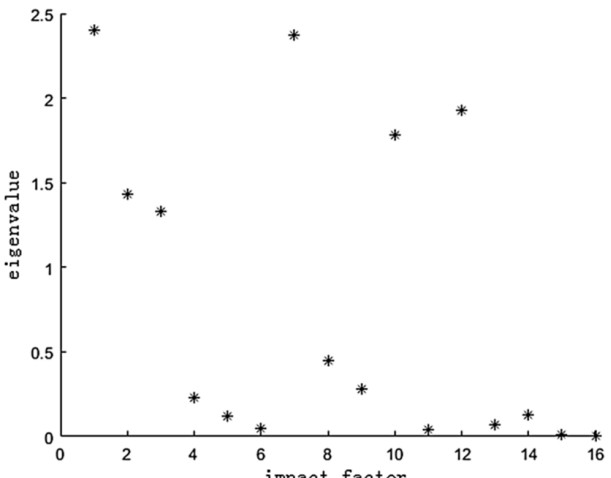

**Figure 7.** Eigenvalue of influence factor.

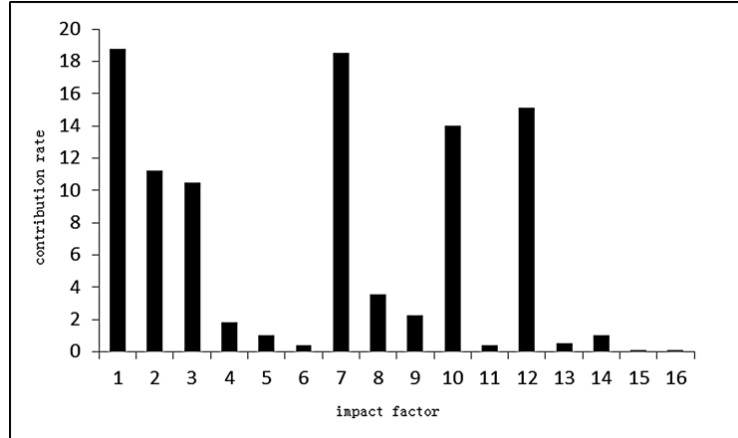

**Figure 8.** Contribution rate of impact factor.

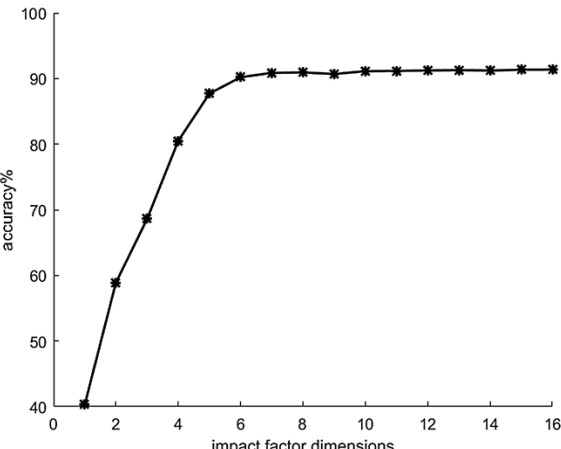

**Figure 9.** Relationship between influence factor dimensions and forecast accuracy.

**Table 2.** Significance and classification of influence factors.

| Influence Factors | Representation Significance | The Probability of Debris Flow/% | | | | |
|---|---|---|---|---|---|---|
| | | <20 | 20~40 | 40~60 | 60~80 | >80 |
| Rainfall/mm | Water source conditions required for debris flow formation, expressed by comprehensive daily rainfall | <20 | 20~50 | 50~75 | 75~100 | >100 |
| Slope gradient/% | The potential energy conditions required for the collection of loose solid matter to quantify the ratio of 25° to 45° area to the basin area | <0.05 | 0.05~0.1 | 0.1~0.15 | 0.15~0.2 | >0.2 |
| Gully bed gradient/‰ | Potential Energy Conditions for Debris Flow Formation | <50 | 50~100 | 100~200 | 200~400 | >400 |
| Relative height difference/m | Terrain Factors of Debris Flow Formation | <20 | 20~40 | 40~60 | 60~80 | >80 |
| Soil moisture content/% | The amount of water in the soil | <8 | 8~16 | 16~24 | 24~32 | >32 |
| Pore water pressure/KPa | The pressure of groundwater in soil or rock, acting between particles or pores. | <10 | 10~25 | 25~40 | 40~60 | >60 |

In this paper, the historical data of 12 monitoring stations in Shanyang County, Shaanxi Province are taken as samples, and 2000 groups of data are selected from the historical data of debris flow in Shanyang County. Among them, 1800 groups and 1900 groups of data are used as training samples, and the remaining data are used as verification samples to train and verify the model. Part of the data collected by the sensor is shown in Table 3.

**Table 3.** Initial impact factor data.

| Number | Impact Factor | | |
|---|---|---|---|
| | Rainfall | Soil Moisture Content | Pore Water Pressure |
| 1 | 18.32 | 6 | 0.20 |
| 2 | 23.28 | 13 | 0.13 |
| 3 | 17.62 | 20 | 0.48 |
| 4 | 38.74 | 27 | −0.26 |
| … | … | … | … |
| 2000 | 138.62 | 33 | 0.74 |

*4.3. Data Preprocessing*

Due to the influence of environmental factors, the sensors used for debris flow monitoring will inevitably have data missing, outliers and inconsistency, so it is necessary to preprocess the data.

(1) Missing value processing

Data missing or outliers is a common situation. The missing sensor data is counted according to its attributes to get the disappeared rate $q$. When $q \geq 90\%$, the column data will be eliminated. When $40\% \leq q < 90\%$, the adjacent attributes are weighted to fill in. When $20\% \leq q < 40\%$, the mean value is used as the filling value, and when $q < 20\%$, the mode is used for filling.

(2) Outlier processing

Some values more than three times the mean distance or five times the standard deviation are outliers and need to be eliminated.

(3) Normalization

The Data we required is not only diverse but also massive. Different types of data have different dimensions. The diversity of data will lead to data imbalance, which will significantly affect the accuracy of the prediction model. The data can be normalized by Equation (18).

$$R' = (R - R_{\min}) / (R_{max} - R_{\min}) \tag{18}$$

where the $R'$ is the normalized data, $R$ is the raw data, $R_{max}$ is the maximum values and $R_{\min}$ is the minimum values.

The data before and after preprocessing are simulated using the matrix random approximation SVDBL model. The grid search is performed on the model parameters to ultimately determine the number of mapped node neurons $N_f = 150$, the number of incremental neurons $N_m = 31$, and the number of incremental neurons $N_e = 23529$.

In addition, the maximum number of iterations $K$ equals 10000, the allowable error $\varepsilon$ equals 0.1, and the SVD oversampling parameter $p$ equals 2. The comparison of the results before and after data preprocessing is shown in Figure 9 and Table 3.

It should be noted that the simulation in this paper is carried out on the CPU2.7ghz, 8G memory machine using MATLAB.

It can be seen from Figure 10 and Table 4 that for the data without preprocessing, when the number of incremental nodes of the model increases to about 7000, the prediction accuracy rate gradually becomes stable, and finally reaches 80%, and the training time is 5.8124 s. However, for the preprocessed data, when the number of incremental nodes of the model comes to more than 2000, the prediction accuracy rate has become stable, reaching 93%, and the training time is 1.2216 s. The simulation results show that it is necessary to preprocess the data, and the data preprocessing method described in this section can effectively improve prediction accuracy.

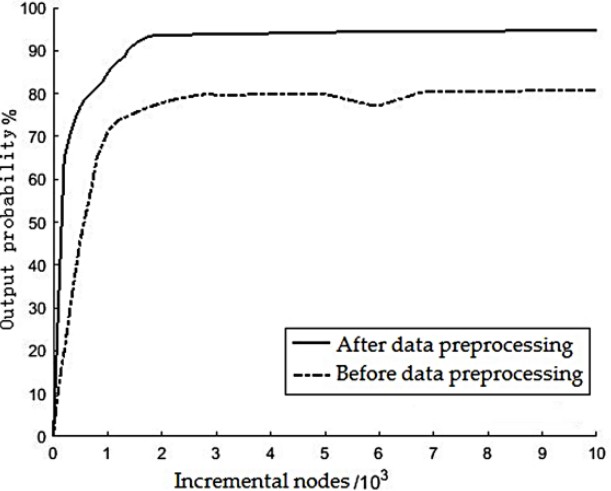

**Figure 10.** Comparison of prediction results before and after data preprocessing.

**Table 4.** Comparison of prediction results before and after data preprocessing.

| Before Data Preprocessing | | | After Data Preprocessing | | |
|---|---|---|---|---|---|
| Incremental nodes/Pcs | Accuracy/% | Training time/s | Incremental nodes/Pcs | Accuracy/% | Training time/s |
| 7000 | 80 | 5.8124 | 2000 | 93 | 1.2216 |

### 4.4. Evaluating Indicator

To further evaluate the performance of the model, the model was tested with root mean square error RMSE, average absolute percentage error MAPE, determination coefficient $R^2$, AUC value and training and test time as indicators.

(1) Root mean square error

$$RMSE = \sqrt{\frac{1}{n}\sum_{i=1}^{n}(y(i) - \overline{y}(i))^2} \tag{19}$$

(2) Mean absolute percentage error

$$MAPE = \sum_{i=1}^{n}\left|\frac{y(i) - \overline{y}(i)}{y(i)}\right| \times \frac{100}{n} \tag{20}$$

where the $y(i)$ is the actual value and $\overline{y}(i)$ is the predicted value of the model.

(3) coefficient of determination $R^2$

$$R^2 = \frac{SSR}{SST} \tag{21}$$

where the SSR is the sum of squares of regression, and SST is the sum of squares of total deviation.

(4) AUC

$$AUC = \frac{\sum rankk_i - \frac{M(M+1)}{2}}{M \times N} \tag{22}$$

where the $M$ is the number of positive samples, $N$ is the number of negative samples and $rankk_i$ is the $i$th sample serial number.

### 4.5. Simulation Verification and the Result Analysis

In the data module, the historical data is retrieved from the server, and the influence factors are extracted in the technical module and calculated in the algorithm module. Finally, the results are obtained in the prediction module, and the model is verified by the right-angle intersection test method. The model's performance is evaluated from the training and testing processes. During the training phase, the gradient descent optimized the BP neural network (GD-BP) model, the grid search support vector machine (SVM) model, the broad learning (BL) model and the matrix stochastic approximation SVD optimized broad learning (SVDBL) model are sequentially trained using the data in Section 3.2. The performance of each model during the training phase is evaluated using the root mean square error RMSE, the mean absolute percentage error MAPE and the goodness of fit R2, respectively. The results are shown in Table 5.

**Table 5.** Comparison of RMSE, MAPE and $R^2$ for models.

| Model | Evaluating Indicators | | |
|---|---|---|---|
| | RMSE | MAPE | $R^2$ |
| GD-BP | 0.3706 | 0.0526 | 0.7826 |
| SVM | 0.3215 | 0.0468 | 0.7583 |
| BL | 0.2541 | 0.0022 | 0.8294 |
| SVDBL | 0.2397 | 0.0212 | 0.8603 |

It can be seen from Table 5 that SVDBL performs best on RMSE, MAPE and $R^2$ in the performance evaluation of the four models. The RMSE of SVDBL is 0.1309, 0.0818 and 0.0144, smaller than that of GD-BP, SVM and BL, respectively. MAPE of SVDBL is 0.0314, 0.0256 and 0.0190 smaller than that of GD-BP, SVM and BL, respectively. Regarding goodness of fit R2, SVDBL is 0.0777, 0.1020 and 0.0309 higher than the above three models, respectively. This shows that the proposed SVDBL model has the best performance.

In the model test phase, the average prediction accuracy and training time of the above four models are compared, as shown in Table 6. The forecast results of each model in the validation set are shown in Figure 11.

**Table 6.** Comparison of average prediction accuracy and average training time of each model.

| Model | GD-BP | SVM | BL | SVDBL |
|---|---|---|---|---|
| Average accuracy/% | 85.5 | 90 | 92.5 | 93 |
| Average training time/s | 23.2906 | 18.1740 | 2.8899 | 1.2216 |

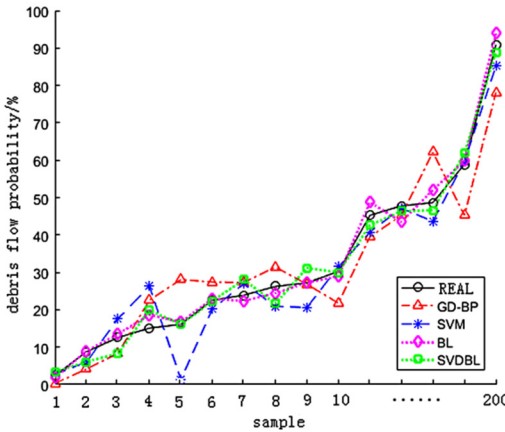

**Figure 11.** Forecast results of four models.

It is not difficult to see from Table 6 and Figure 11 that the SVDBL is 7.5% higher than GD-BP, 3% higher than the SVM and 0.5% higher than the BL in forecasting accuracy. Regarding training time, the SVDBL is 22.0690 s faster than the GD-BP, 16.9524 s faster than the SVM and 1.6683 s more quickly than the BL. Therefore, the SVDBL can correctly predict the probability of debris flow disaster, and the SVDBL has more obvious advantages than other models in training time.

To further verify the model's performance, the four models' prediction effects are compared by RMSE and $R^2$. The results are shown in Figures 12 and 13.

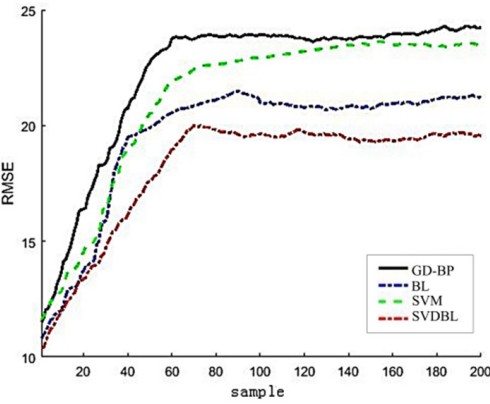

**Figure 12.** RMSE comparison diagram of root mean square error of four models.

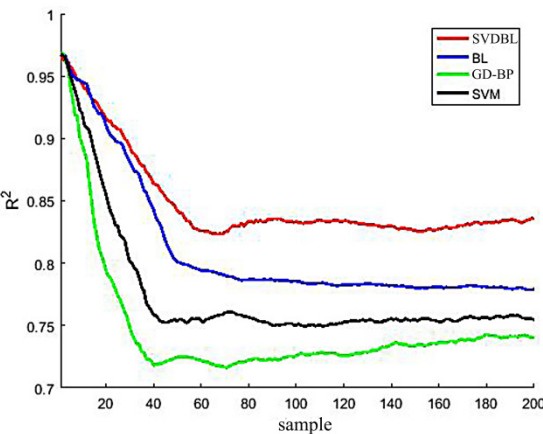

**Figure 13.** Comparison of determination coefficient $R^2$ of four models.

It can be seen from the curve of Figure 12 that the RMSE shows a steep upward trend when there are few samples and finally tends to be stable after a period of small oscillation. The overall variation trend of the RMSE of the four models is similar, but the stable root means a square error of the BLSVD is only 0.1876, which is 0.0587, 0.0478 and 0.0227 smaller than the stable RMSE of GD-BP, SVM and BL, respectively, which shows that the model has the best performance.

From the comparison of the curve of Figure 13, it can be seen that the $R^2$ of the overall model first shows a decreasing trend and a slow and stable trend. The $R^2$ of GD-BP is basically below 0.75. The support vector machine SVM is superior to GD-BP, and the $R^2$ is slightly higher than 0.75. The $R^2$ of BL is between 0.75 and 0.8, and the performance is somewhat outstanding for the support vector machine SVM. The SVDBL is better than other models, and the $R^2$ is maintained above 0.80, indicating that the model works best.

Then the model's performance is compared with the mean absolute percentage error MAPE. After calculation, the MAPE of GD-BP, SVM, BL and SVDBL models are 3.22%, 2.93%, 1.76% and 1.27%, respectively. SVDBL has the smallest MAPE and the best prediction.

To enable the management personnel to specify the emergency plan in time according to the obtained probability value of debris flow, the rules of debris flow early warning level are given below, as shown in Table 7. According to the "Land and Earthquake Prediction and Geological Disaster Warning Agreement", jointly carried out by the Ministry of Land and Resources and the Meteorological Bureau of China, the debris flow warning level is divided into five groups. Different warning levels correspond to different risk, color identification, interpretation and probability values. In other words, it means that after predicting the probability value, the early-warning level of debris flow disaster can be obtained.

**Table 7.** Early warning classification of debris flow geological disasters.

| Warning Level | Possibility of Disaster | Color Identification | Explanation | Probability |
|---|---|---|---|---|
| I | Very low | Green | Only send information to decision-makers | <20% |
| II | Lower | Blue | Push information to decision-makers and related technical personnel | 20%~40% |
| III | Medium | Huang | Recommend preventive measures | 40%~60% |
| IV | Higher | Orange | Take preventive measures | 60%~80% |
| V | Extremely high | Red | Organizing a public emergency response | >80% |

In addition, the model is verified by the Receiver Operating Characteristic (ROC) curve, as shown in Figure 14 and Table 8. The average AUC value of the BL model is 90.37%, which is 12.75% and 7.64% higher than the GD-BP and SVM models. Compared with the BL model, The AUC value of the optimized SVDBL model is 2.79% higher than that of the BL. The test time of SVDBL is only 0.0010 s, which is 0.0807 s, 0.0039 s and 0.0012

s shorter than GD-BP, SVM and BL, respectively. Therefore, applying the SVDBL model to debris flow forecast is feasible and has a good forecast effect and running speed.

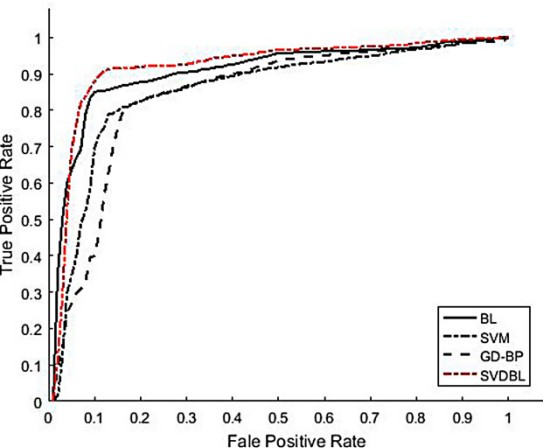

**Figure 14.** Average ROC curves of four models.

**Table 8.** Comparison of average ROC curves.

| Model | Average AUC Value/% | Average Training Time/s |
| --- | --- | --- |
| GD-BP | 80.41% | 0.0817 |
| SVM | 85.52% | 0.0049 |
| BL | 90.37% | 0.0022 |
| SVDBL | 93.16% | 0.0010 |

To verify the online training ability of the SVDBL model, the training samples were expanded to 1900 groups based on the original 1800 groups of data. The models were retrained and the results were compared (Table 9). The comparison results in Table 9 show that after the expansion of training samples, the average training time of the GD-BP model, the SVM model and the BL model increased by 8.6652 s, 4.8129 s, and 1.1674 s, respectively, while the average training time of the SVDBL only increased by 0.0084 s. After the training sample is expanded, the average training time of the SVDBL model hardly changes, which shows that the optimized SVDBL model has a robust online training ability. The above conclusion is only the result of the sample expansion by 100 datasets. When the scale of the training sample is further expanded, the advantages of the SVDBL will be more obvious.

**Table 9.** Comparison of results before and after training sample expansion.

| Model | Results of before Training Sample Expansion | Results of after Training Sample Expansion |
| --- | --- | --- |
| | Training Time/s | Training Time/s |
| GD-BP | 23.2906 | 31.9558 |
| SVM | 18.1740 | 22.9869 |
| BL | 2.8899 | 4.0573 |
| SVDBL | 1.2216 | 1.2299 |

To verify the effectiveness of introducing the FMPCE algorithm into the model proposed in this paper, four groups of different influence factors are selected artificially (shown in Table 10, and the influence factors represented by serial numbers in the table are shown in Section 3.2). The prediction results of the model are verified by simulation, and some results are shown in Figure 15.

**Table 10.** Correspondence between the influence factor and the curve in Figure 7.

| Artificially Selected Impact Factors | The Corresponding Curve in Figure 7 |
|---|---|
| 1,2,3,7,12,14 | b |
| 1,2,3,4,5,12 | c |
| 1,2,7,8,10,12 | d |
| 1,3,6,7,10,14 | e |

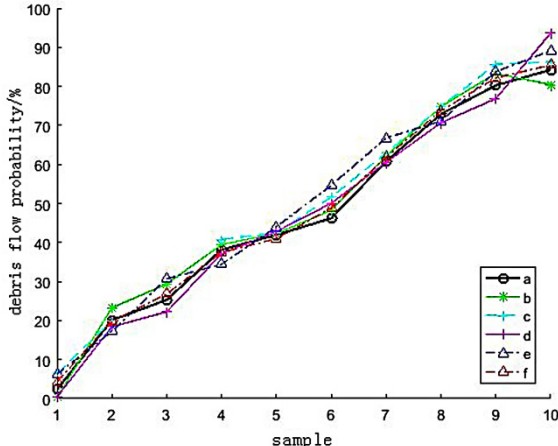

**Figure 15.** Comparison of prediction results of different influence factors.

In Figure 15, curve a shows the actual results, curve f shows the forecast results of the influence factors extracted by the FMPCE algorithm and the influence factors corresponding to other forecast results curves are shown in Table 7. Compared with curves b, c, d and e in Figure 15, curve f is closer to curve a, which means the influence factor extracted by the FMPCE algorithm is more relative to the actual value than the predicted result of artificially selecting the influence factor, which also indicates that the method of removing the influence factor adopted in this paper can overcome the influence of people's subjective factors, and is more effective than the method of only manually selecting the influence factor.

## 5. Conclusions

This paper constructs a debris flow disaster prediction model based on the FMPCE algorithm and width learning. The model is optimized by matrix random approximation singular value decomposition SVD, and the SVDBL model is established to predict the probability of debris flow. The relationship between debris flow factors and disaster probability is analyzed and verified by simulation. At the same time, the model adopted in this paper is compared with the GD-BP prediction model, the SVD prediction model and the BL prediction model.

(1) According to the influence of different dimensions on the accuracy of model prediction, the main impression factors for its occurrence probability in the process of debris flow formation, are closely related to the rainfall, soil moisture content and pore water pressure collected by sensors, besides the common topographic factors such as slope gradient, gully bed gradient and relative height difference.

(2) Debris flow is a complex system with remarkable nonlinear characteristics, formed under rainfall, geographical, human, and other factors. Therefore, the method of combining the FMPCE algorithm with the BL model is applied to the prediction model of the debris flow. Compared with the previous methods such as neural network, analytic hierarchy process and fuzzy extension theory, it can not only reveal the complexity and fuzziness of each influencing factor, but also objectively reflect the relationship between each influencing factor and debris flow, and minimize the influence of subjective factors. In prediction

accuracy, the model processed by FMPCE is 13% higher than the model not processed by FMPCE. From the perspective of the number of incremental nodes, the model processed by FMPCE saves about 5000 enhanced node costs than the model not processed by FMPCE.

(3) The input matrix of the hidden layer of the BL is optimized by the matrix random approximation SVD, which effectively avoids the redundancy of matrix structure caused by random initialization, dramatically improves the training speed of the debris flow prediction model, and provides effective support for online training of debris flow warning system in practical application.

(4) The model is verified by the sample set. Under a limited number of samples, from the average prediction accuracy, the optimized model is 0.5% higher than the unoptimized model, and the average AUC value is increased by 2.79%. In terms of online updating performance, the training time is shortened by 0.69%, which significantly improves the training speed of the debris flow forecasting model. The high accuracy, the AUC value, and the fast training and forecasting speed, all indicate that applying this model to the probability forecast of debris flow is feasible. A new idea is provided for the practical application of debris flow prediction.

(5) In addition to predicting the occurrence probability of debris flow disaster, when the disaster occurs is also a concern of many researchers. However, broad learning cannot deal with the dynamic characteristics of the system. It is difficult to make a reasonable prediction of time. In the future, we can consider combining the structure of the recurrent neural networks and echo state networks with better dynamic capture ability with width learning to improve the time prediction ability of the model.

**Author Contributions:** Writing—original draft, G.X. (Genqi Xu); writing—review & editing, X.-E.Y. and G.X. (Guokun Xie); resources N.C., G.X. (Guokun Xie), J.M. and L.L.; data curation, J.M.; formal analysis, L.L. All authors have read and agreed to the published version of the manuscript.

**Funding:** This research was funded by the Key Scientific Research Projects of Xi'an Traffic Engineering Institute grant number (2021KY-25), the Scientific Research Plan Projects of Shaanxi Education Department grant number (2022JK0515).

**Data Availability Statement:** Data were obtained from 12 key geological hazard monitoring sites in Shanyang County, Shaanxi Province, China.

**Acknowledgments:** This work was financially supported by The Youth Innovation Team of Shaanxi Universities in 2022 and Key Scientific Research Project of Xi'an Traffic Engineering Institute (2021KY-25). Papers, works, academic reports and other achievements published by the innovation team should be labeled in Chinese "陕西高校青年创新团队", "The Youth Innovation Team of Shaanxi Universities" in English.

**Conflicts of Interest:** The authors declare no conflict of interest.

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
