# Peer review of "Debris Flow Prediction Based on the Fast Multiple Principal Component Extraction and Optimized Broad Learning"

_water, doi:10.3390/w14213374_

Round 1

Reviewer 1 Report

The manuscript is about the prediction of debris flow probability using machine learning models. It is within the scope of “Water”. I have following comments for the presented draft towards improving.

1.     Please ensure that all the keywords are also mentioned within the Abstract.

2.     I can see that most of the graphs are not of the vector graphics, it is recommended to include the high quality (vector graphics if available) images for the curves.

3.     First, authors need to highlight the intuition behind the research performed in this manuscript. Somewhere, in introduction, it is recommended to add information about why this research is needed? What research gap is this research trying to address? Why only specific machine learning models used? 

4.     I will also recommend to list the contributions of manuscript clearly at the end of introduction section.

5.     I have a fundamental concern related to the problem type, is this considered as a regression problem or a classification problem? Authors are requested to clearly highlight the scope of research in the methods section and introduction.

6.     Why only the mentioned methods? Now, given that, deep learning models are very popular and have achieved so much success, why not ANN or CNN were used for this problem? Also, some high performing conventional models like XGBoost was also not implemented. Authors need to justify their choices.

7.     I would suggest the authors to have a separate section about the dataset. About the dataset statistics, what number of sample, a head of dataset if possible and description of what each feature in the dataset represents. It would be nice if authors may choose to give the developed dataset a unique name. 

8.     I would suggest to add a separate section about the experimental protocols and evaluation measures. Authors may add information about what platform was used for training, what hyperparameters for the implemented models were used, what dataset split was used, and what different type of evaluation measures were used.

9.     An important aspect of practical implementation is missing in the article. For me, having no implication section discussing a potential real-world use case is one of the significant lacking of this manuscript. I would suggest authors to add a use-case in which, the proposed research can be deployed as an end-to-end solution. Description of such a practical system will be valuable contribution for disaster management agencies, government officials and policymakers. 

10.  Please clearly highlight the limitations of the presented research and highlight the future research directions.

Reviewer 2 Report

The paper needs significant revisions:

(1) Quantitative performance of ML models should be mentioned in the abstract section.

(2) Introduction section has no motivation and innovation. Additionally, a robust research organization is needed. 

(3) Literature review can be improved by using relevant references:

-Neuro-fuzzy GMDH-based evolutionary algorithms to predict flow discharge in straight compound channels

-Optimized expressions to evaluate the flow discharge in main channels and floodplains using evolutionary computing and model classification

(4) setting parameters of machine learning models should be mentioned and clarified.

(5)The performance of training and testing stages should be expanded using R, RMSE, and MAPE.

Round 2

Reviewer 1 Report

I thank the authors to take into consideration most of my comments. The manuscript is in much better shape now. I am satisfied with the modifications made. Just one inconsistency is observed regarding the heading numbers and corresponding sub-heading numbers. I can see sub-heading 3.XX under Section 4. Also Conclusion should be section 5 and not section 4. 

Reviewer 2 Report

Accept as is